# RasGRP2 Structure, Function and Genetic Variants in Platelet Pathophysiology

**DOI:** 10.3390/ijms21031075

**Published:** 2020-02-06

**Authors:** Matthias Canault, Marie-Christine Alessi

**Affiliations:** 1Aix Marseille University, INSERM, INRAE, C2VN, 13005 Marseille, France; 2Hematology laboratory, APHM, CHU Timone, 13005 Marseille, France

**Keywords:** platelet, RasGRP2, inherited platelet disorder

## Abstract

RasGRP2 is calcium and diacylglycerol-regulated guanine nucleotide exchange factor I that activates Rap1, which is an essential signaling-knot in “inside-out” αIIbβ3 integrin activation in platelets. Inherited platelet function disorder caused by variants of *RASGRP2* represents a new congenital bleeding disorder referred to as platelet-type bleeding disorder-18 (BDPLT18). We review here the structure of RasGRP2 and its functions in the pathophysiology of platelets and of the other cellular types that express it. We will also examine the different pathogenic variants reported so far as well as strategies for the diagnosis and management of patients with BDPLT18.

## 1. RasGRPs General Description

### Cell Signal Transduction Is a Finely Regulated Process That Relies on Several Control Hubs

Members of the RAS family of small guanosine triphosphatases (GTPases; including RAS and Rap) are among the essential regulators of cell signaling. They are acting as binary molecular switches that transmit signal when bound to GTP [1]. Their activation is mediated by guanine-exchange factors (GEFs) that facilitate guanosine diphosphate (GDP)-dissociation and its replacement by GTP [2]. Signal termination is mediated through the action of GTPases-activating proteins (GAPs) [2] that catalyze the hydrolysis of the bound GTP to GDP by increasing the relatively slow intrinsic catalytic activity of the GTPase by ≈10^5^ fold [3].

Several families of GEF are described in human cells [4] such as the Son of Sevenless, that activates Ras [5], Epac1 and Epac2, that activate Rap [6] and the Ras guanine-nucleotide releasing proteins (RasGRPs) that act on both Ras and Rap. The RasGRP family consists in four different forms (RasGRP1-4). Their expressions are not ubiquitous but can be overlapping and are rather restricted to specific cell types and tissues, mainly brain, vascular and hematopoietic cells. RasGRP1 expression concentrates essentially to the cerebellum, the cerebral cortex and the amygdala [7] as well as T cells. It is also present in B-, natural killer (NK)- and mast cells but to a lesser extent [8,9,10]. RasGRP2 expression was initially described in striatum neurons [7] and was then found in platelets and their precursors, the megakaryocytes as well as neutrophils [11,12,13]. It has now been detected in other hematopoietic-lineage-derived cells such as T lymphocytes [14] but also found in fibroblast-like synoviocytes [15] and endothelial cells [16]. RasGRP3 is highly expressed in B cells [17], [18] but is also expressed in T cells, macrophages, and endothelial cells [17,18,19,20]. RasGRP4 is relatively specific to mast cells, although it has also been detected in thymocytes and neutrophils [21,22,23] in synoviocytes [15].

## 2. RasGRPs Structure and Domain Organization

RasGRP1-4 are multidomain proteins sharing common structural organization (Figure 1) with a high degree of sequence identity [24]. They contain a central catalytic module composed of two domains: 1) the cell division cycle-25 (CDC-25) homology domain that directly interacts with GTPase and encloses an helical hairpin that dislodges the bound GDP nucleotide from the GTPase; and 2) the Ras exchanger motif (REM) domain, that is essential for the catalytic activity of the RasGRPs through providing structural support for the CDC-25 domain [5] but that is not conserved among all Ras-GEFs. Besides the catalytic core, RasGRPs are composed of a second shared module that consists in a pair of successive helix(E)-loop-helix(F) (EF)-hand domains, with each one being capable of binding one calcium ion. EF hands are typical helix-loop-helix structural motifs that coordinate calcium via acidic residues within the loop. Calcium binding results in important conformational changes within the domain and can also lead to modifications of the structure of other domains of the protein [25]. The remaining part of the RasGRPs consists of a C1 domain that was originally described as a diacylglycerol (DAG) binding cassette on protein kinases C (PKCs), responsible for their membrane localization. The “atypical” C1 domains of RasGRP2 stands out as it binds DAG with a very weak affinity and that DAG (or its analogs, phorbol esters) does not drive RasGRP2/membrane association as for the C1 domains of the other RasGRPs [26].

Finally, RasGRP1 in the only RasGRP family member that has a C-terminal sequence predicted to form a coiled-coil domain that enhances membrane recruitment through electrostatic interactions with phosphoinositides [24].

## 3. RasGRP2

RasGRP2, calcium and diacyglycerol-regulated guanine exchange factor I (CalDAG-GEFI, UniPortKB: Q7LDG7-1) is encoded by the *RASGRP2* gene (Ensembl: ENSG00000068831) that is located on chromosome 11 (11q13.1 locus, gene/locus MIM#605577), spans 18.55 kb, and contains 22 exons. Four different isoforms that are produced by alternative splicing (Q7LDG7-1 to -4) have been described so far and nine other potential isoforms have been computationally mapped. *RASGRP2* gene expression is under the control of the transcription factor NE-F2 [27] that was shown to regulate late phase of megakaryocytic differentiation [28] and to be required for proper αIIbβ3 integrin activation and fibrinogen binding [29], key steps in platelet aggregation. Indeed, RasGRP2 expression raises in late stages of megakaryocytic lineage differentiation during polyploidization [30] assuring the presence of sufficient amounts of RasGRP2 in produced platelets.

The canonical isoform 1 of RasGRP2 (Q7LDG7-1) is a 609-amino-acid-long (69.25 kDa) protein that possesses various post-translational modification sites that were identified through high-throughput proteomic analyses (data obtained from PhosphoSitePlus [31]) and could affect the activity and/or the fate of the GEF. Nine serine-, two threonine-, and one tyrosine-phosphorylation sites were identified. Besides, 10 putative ubiquitination lysine residues and one myristylation site were annotated. Among those, four serine-phospho sites were validated using methods other than discovery mass spectrometry and their implication in the regulation of RasGRP2 activity were further characterized (see chapter RasGRP2 activity regulation).

## 4. RasGRP2 Functions in Platelets

RasGRP2 diverges from the other members of the RasGRP family as it catalyzes GDP to GTP exchange only for Rap GTPases but not Ras [7]. Rap1 is a ubiquitous protein that plays an essential role in the control of many cellular processes such as cell division, adhesion, and cell migration [32]. In platelets, the most abundant Rap GTPases are Rap1A and B with 125,000 and 300,000 copies/platelet respectively [33] that exhibit functional redundancy [34]. Several Rap GEFs have been detected in platelets such as RasGRP3 [35], PDZ-GEF1 [35] and Epac1 [36], but to date only RasGRP2 was shown to be implicated in platelet function regulation. The initial demonstration of RasGRP2 involvement in platelet function essentially comes from studies performed in mice. Work from the Shattil group in the early 2000s demonstrated that in mouse embryonic stem cell derived megakaryocytes, the retroviral overexpression of RasGRP2 leads to enhanced agonist-induced activation of Rap1 and fibrinogen binding to the αIIbβ3 integrin [27]. Then, using the *Rasgrp2* deficient mice developed by Crittenden and coworkers, the role in vivo of RasGRP2 in Rap1 and in αIIbβ3 integrin “inside-out” activation processes in platelets was unequivocally established [11]. Further work on platelets from these mice led to establish the molecular mechanisms linking RasGRP2/Rap1 and the two pathway models for platelet activation:

Platelet surface receptor activation by most agonists initiate intracellular signaling pathways through the phospholipase C isoforms β or γ (depending on the class of surface receptor enrolled) which hydrolyze phosphoinositide-4,5-bisphosphate (PIP2) to inositol-1,4,5-trisphosphate (IP3) and 1,2-diacyl-glycerol (DAG). IP3 induces the release of Ca2+ from intracellular stores into the platelet cytoplasm [37,38] and DAG activates protein kinases C (PKCs) that results in platelet sustained granule secretion, subsequent adenosine diphosphate (ADP) release and P2Y12 receptor activation. These Ca2+-sensitive and PKC pathways were shown to act separately but synergistically in the activation of αIIbβ3 integrin [39,40].

Studies with murine RasGRP2–deficient platelets demonstrate that the GEF is predominantly regulated by Ca2+ signals and its involvement in integrin activation is independent of the PKC/P2Y12 pathway. Indeed, RasGRP2 is critical for the rapid, but reversible, activation of Rap1 as observed upon low dose thrombin activation, that is dependent on the increase of cytoplasmic Ca2+ concentration [11,12]. The second pathway is RasGRP2-independent and leads to slower but sustained Rap1 activation [41]. It involves PKC signaling [12,42], ADP secretion and P2Y12-dependent [41,43,44] activation of PI3K [43,45] that causes inhibition of RASA3 (GAP1IP4BP), the most abundant Rap1 GAP found in platelets [33,46,47]. RASA3 is required to maintain circulating platelets in a quiescent state through antagonization of low-level Rap1 activation and its inhibition prevents GTP hydrolysis from Rap1-GTP and thus enables substantial platelet activation [48]. This two-pathway model of platelet activation (Ca2+/RasGRP2 and P2Y12/RASA3) is a balance tightly regulated by several activator and inhibitory signals (see for review [49,50,51]) that converge to Rap1 activation and downstream favoring αIIbβ3 integrin activation [52].

Consistent with the involvement of RasGRP2 in αIIbβ3 integrin activation, mouse platelets lacking the GEF show a markedly reduced ability to form three-dimensional thrombi when perfused at arterial shear rates both in vitro and in vivo and dramatically prolonged bleeding time [11,41]. RasGRP2-deficient platelets have impaired aggregation in response to any dose of calcium ionophore (A23187) and weak agonists (ADP and the thromboxane A2 analog, U46619) and to low doses of strong ones such as thrombin and collagen [11]. Interestingly, hypomorphic mice expressing minimal levels of human RasGRP2 instead of endogenous RasGRP2 (≈10% of expression in controls) showed reduced platelet aggregation and severely impaired arterial and immune complex-mediated thrombosis with only slightly affected primary hemostasis [53]. Thus, RasGRP2 could represent a therapeutic target for the development of potentially safe antithrombotic drugs with little impact on the bleeding risk.

In parallel, RasGRP2 also contributes to thromboxane A2 generation and release from mouse platelets, thus reinforcing the second wave of platelet activation signal through PKC-mediated ADP secretion and the P2Y12/PI3K-dependent RASA3 inhibition pathway [48,54].

Additionally to integrin “inside-out” activation, clues exist for the possible involvement of RasGRP2 in platelet spreading and integrin “outside-in” signaling. In mice, the loss of Rap1b inhibits platelet spreading over fibrinogen [55] and mouse platelet lacking both RasGRP2 and P2Y12 receptor fail to spread over fibrinogen [56]. Additionally, platelets from RasGRP2 knock-out mice show impaired α2β1 integrin-dependent spreading over collagen [57]. Then, Stefanini and colleagues demonstrated that RasGRP2-deficiency in murine platelets resulted in altered activation of Rac1, the RhoA family GTPase that controls lamellipodia extension and subsequently platelet spreading [56]. In humans, we reported defective spreading over fibrinogen of platelets from patients expressing an inactive form of RasGRP2 [58]. However, other studies on platelets isolated from individuals lacking the GEF showed only minimal spreading impairment [59,60]. Whereas the effect of RasGRP2 deficiency on platelet spreading relies on consequences of defective integrin “inside-out” activation or on its direct involvement in “outside-out” signaling remains to be fully determined.

The signaling module RasGRP2/Rap1 was also shown to play an important role in the conversion of platelets to a pro-coagulant state since RasGRP2-deficiency leads to impaired phosphatidylserine exposure on mouse platelet surface and delayed and reduced fibrin generation at the vascular lesion site [61]. However, the exact molecular mechanism linking activated Rap1-GTP and phosphatidylserine exposure in platelets still remains to be elucidated.

A role for RasGRP2 was also proposed in atherogenesis. Indeed, RasGRP2/Rap1-dependent signal promotes atherosclerotic plaque formation in mice and determines its composition probably through platelet activation and platelet-leukocyte aggregate formation. However, RasGRP2 is also expressed in leukocytes, thus the exact contribution of platelet- and/or leukocyte-associated RasGRP2-dependent signal in atherosclerosis still remains elusive [62].

## 5. RasGRP2 Functions Outside Platelets

RasGRP2 is expressed by developing and mature neutrophils [11]. It has been then involved in in vitro and in vivo chemotaxis, adhesion and extravasation in a manner that either depends on integrin or on mechanisms involving E-selectin [13,63,64]. Similarly, RasGRP2 enhances the adhesion ability of human T cells through lymphocyte function-associated antigen-1 (LFA1) and contributes to the interaction with intercellular adhesion molecule-1 (ICAM-1) [14]. In human T cells, the translocation of RasGRP2 to the cell membrane via interaction with polymerized actin was observed in response to TCR stimulation [65] where the GEF co-localizes with its substrate, Rap1 [66]. However, the exact contribution of RasGRP2 to leukocyte function in vivo remains a matter of debate because no evidence of immune disorder has been reported in the human cases of RasGRP2 deficiency described so far.

Additionally, several groups reported that RasGRP2 might be involved in oncohematological diseases. RasGRP2 has been identified as the proto-oncogene in acute myelogenous leukemia and its expression was found to be increased in trisomy 12-associated chronic lymphocytic where it is thought to contribute to the drug resistance-associated enhanced integrin signaling [67,68]. Additionally, the RasGRP2/Rap1 axis mediates the chronic lymphocytic leukemia cell adhesion and migration in response to an increase in intracellular Ca2+ levels upon CD38 engagement [69].

Regarding RasGRP2 and endothelium, a combination of microarray and expression pattern analysis, allowed to identify xrasgrp2 in *Xenopus* embryos as a vascular-expressed gene, that is a homolog to the human form of *RASGRP2*. It has then been further involved in vasculogenesis and/or angiogenesis during *Xenopus* embryo development [70]. The endothelial expression of RasGRP2 was also observed in human vascular cells from both venous and arterial origins [16]. Recently, RasGRP2 was suggested to contribute to maintenance of endothelial homeostasis, as its overexpression in human umbilical vein endothelial cells (HUVECs) prevents TNF-α-induced ROS production and apoptosis via a Rap1 activation-dependent mechanism [71]. Additionally, the overexpressed RasGRP2 in HUVECs also suppresses apoptosis induced through Bax-activation via a Rap1-independent but R-Ras-dependent signaling pathway [72]. Furthermore, the RasGRP2 protein was found to be abundantly increased in the vascular endothelium and in fibroblast-like synoviocytes retrieved from synovial tissues of a subset of patients suffering rheumatoid arthritis [15]. In synoviocytes, RasGRP2 expression is induced in response to growth factors (i.e., platelet derived growth factor, PDGF and vascular endothelial growth factor, VEGF) and transforming growth factor-beta (TGF-β). It controls, through Rap1 activation, both actin-dependent adhesion/migration and interleukin-6 production via a NF-κB-mediated pathway [15].

RasGRP2 was initially identified as a protein with enriched expression in human and rodent brain basal ganglia and in their axon-terminal regions [7,73]. Crittenden and colleagues further characterized this localized expression in the striatum area. They also found that RasGRP2 expression was markedly down-regulated in the striatum of patients with Huntington’s disease [74] and in mice with both Huntington’s and Parkinson’s diseases that are major extrapyramidal disorders in which striatal abnormalities are the causes of the pathology [74,75]. More recently, RasGRP2 was involved via Rap1 activation in dopamine-dependent neuronal excitability and reward-related behaviors [76]. Thus, RasGRP2 may be an important regulator of specific behaviors but its function in neurologic functions and human pathologies remains largely to be elucidated.

## 6. RasGRP2 Activity Regulation

RasGRP2 plays critical roles in platelet and other cell type function regulations. Its activity must be finely tuned. The main trigger for the activation of RASGRP2 is calcium; however, other regulation mechanisms have been described such as phosphorylation of serine residues by protein kinase A (PKA) and extracellular signal-regulated kinases 1/2 (ERK1/2) or interaction of its C1 domain with specific membrane phosphoinositides that orientates the GEF towards to a localization to the cell membrane (Figure 2).

### 6.1. Calcium

As mentioned above, RasGRP2 rapidly activates Rap1 in response to an increase in cytoplasmic calcium concentration [54]. RasGRP2 binds to calcium via its two EF hand domains which have a high affinity for calcium (Kd < 100 nM) [77], making RasGRP2 extremely sensitive to activation, since the concentration of cytoplasmic calcium ranges from 25–100 nM in resting platelets and can increase up to micromolar levels upon activation depending on the agonist and the dose [78]. The binding of calcium to the EF hand domains results in major conformational changes [25]. Biochemical and biophysical approaches revealed that calcium binding to EF hands induces global conformational changes in the structure of RasGRP2, most prominently in an auto-inhibitory linker region located between the CDC-25 and the first EF-hand domains, that blocks the catalytic surface of the CDC-25 domain and prevents Rap1 engagement [79].

### 6.2. Phosphorylation

RasGRP2 phosphorylation was also proposed to be control levers to its guanine-exchange activity. Two studies have demonstrated that PKA phosphorylates RasGP2 on multiple sites (Ser116/Ser117 and Ser587) and that phosphorylation correlates with the inhibition of Rap1 activation in platelets [80,81]. Overexpression of phosphor-mimetic or non-phosphorylatable forms of RasGRP2 in HEK297 cells further confirms this inhibitory effect. Most recent data obtained from platelets, have shown that the PKA-dependent phosphorylation of RasGRP2 on Ser587 is clearly downregulated upon ADP stimulation [82]. This might explain, at least partially, the reversible nature of RasGRP2 activation in activated platelets. However, phosphoproteomic study identified RasGRP2 as a substrate for PKA phosphorylation on Ser116, Ser117, Ser554, and Ser587 in striatum neurons promoting its GTP-exchange activity on Rap1 [76]. Thus the regulatory effect of PKA on RasGRP2 activity may have to be considered in regards to the cell type it is expressed in.

Using experimental approaches, ERK1/2 has also been proven to phosphorylate RasGRP2 on Ser394 [83]. In HEK293T cells transfected with a phospho-mimetic variant of RasGRP2 the ERK1/2-dependent phosphorylation of Ser394 impairs RasGRP2 nucleotide exchange activity. This defines a negative-feedback loop that regulates the ERK signaling cascade that is activated downstream of Rap1 in platelets [54].

### 6.3. C1 Domain

The C1 domain provides additional regulatory activity. Structural studies by Iwig et al. demonstrated that DAG binds to the C1 domains of the RASGRP1, 3, and 4 unlike RasGRP2 [77]. As an example, the interaction of DAG with the C1 domain of RasGRP1 releases C1 domain dimerization, subsequently causing the protein membrane translocation. In contrast, RasGRP2 C1 domain is monomeric [77] and RasGRP2 deficiency has no effect on platelets aggregation in response to DAG analogs [12,58]. The physiological importance of the C1 domain of RasGRP2 in platelets is highlighted by the dramatic effect of its loss on platelet function in vitro and in vivo in both mice and humans [41,84]. Lipid co-sedimentation assays and molecular dynamics simulations with cellular localization experiments demonstrate that the atypical C1 regulatory domain of RasGRP2 controls subcellular localization by interacting with the membrane phosphoinositides, phosphatidylinositol (4,5)-biphosphate (PIP2) and phosphatidylinositol (3,4,5)-triphosphate (PIP3) [85]. Specific C1 residues Arg508, Arg513, and Arg530 contribute to PIP2/3 specific binding, facilitating the recruitment of the membrane-associated Rap1 and allowing downstream αIIbβ3 integrin activation.

## 7. RasGRP2 Variants and RasGRP2-Related Bleeding Disorders

### 7.1. In Animals

In 1980, a presumably genetic disorder responsible for bleeding in Simmental cattle was described for the first time [86]. The affected cattle showed spontaneous nosebleeds, hematuria, hematomas and excessive bleeding after injuries or surgery. This bleeding was then demonstrated to be the result of a hereditary thrombopathy likely caused by a defect in calcium mobilization or utilization by platelets [86,87]. In 2007, Boudreaux et al. associated this recessively inherited hemorrhagic disease with a mutation in the *RASGRP2* gene [88]. Sequencing *RASGRP2* in samples from the affected calf revealed a homozygous single-nucleotide change in exon 7 (c.701T>C) that results in the p.Pro234Leu transition (Figure 2). This variant was then considered likely to have an impact on the function of the protein. The same team reported, also in 2007, cases in three different dog breeds (Basset Hounds, Eskimo Spitz and Landseers) of recessively inherited *RASGRP2* mutations (Basset hounds: c.509-511delTCT, p.Phe170del; Eskimo Spitz: c.452dupA, p.Asp151Glufs*115; Landseers: c.982C>T, p.Arg328*, Figure 3) [89]. All affected cases suffered recurrent epistaxis, gingival bleedings, and petechiae. Platelet function was also impaired in these animals as aggregation responses to ADP, collagen, calcium ionophore (A23187), and platelet-activating factor (PAF) was markedly reduced.

### 7.2. In humans

In 2014, our group identified the first pathogenic variant of *RASGRP2* in three siblings affected by platelet related bleeding disorder. [58] Since this first description, 23 other patients have been recorded and this hemorrhagic pathology has been referenced as bleeding disorder-platelet type-18, BDPLT18 (OMIM# 6158888).

### 7.3. Diagnosis

The initial step in the diagnosis of BDPLT18 is to determine the medical history of the patient and its related (parents and siblings). The family history of cutaneous and mucosal bleedings along with their severity and frequency should be documented. Additionally, the presence or not of consanguinity in the patient’s pedigree must be highlighted. Purpura, petechiae, epistaxis, easy bruising, and menorrhagia are common features of the disease and it is mostly, although not always, diagnosed at an early age. Indeed, all reported patients presented abnormal bleedings that were predominantly epistaxis (96%), mucocutaneous bleedings (84%), menorrhagia (73% of affected females), or following dental extraction (40%) or surgery (40%) (Table 1). Gastrointestinal bleeding occurred in four cases (16%). Intracranial bleeding has so far never been reported. Interestingly, as for Glanzmann thrombasthenia (BDPLT16, OMIM# 187800) patients, the incidence of severe bleedings decreases with age [58,59].

Patients affected by BDPLT18 have platelet counts that are typically in the normal range and show normal platelet morphology. The excessive mucocutaneous bleedings are suggestive of severe platelet pathology and after excluding coagulation and von Willebrand factor abnormalities, the main challenge will be to distinguish BDPLT18 from Glanzmann thrombasthenia.

PFA100/200 closure times in response to collagen/ADP or collagen/epinephrine are both prolonged (> 300 sec) [60,95]. Platelet function testing rapidly orientates towards BDPLT18 for it is, until today, the only disorder where platelet aggregation is absent to low dose of agonists (e.g., ADP 5 µM, collagen 2 µg/mL, TRAP10 µM, epinephrine 5 µM) while the response to ristocetin and high doses of agonists (e.g., ADP 20 µM, TRAP 50 µM, collagen 20 µg/mL, arachidonic acid 1.5 mM) or PMA is maintained. This typical profile is nevertheless subject to variability that may be related to the different commercial sources of reagents used or to patient characteristics. This has been illustrated by Sevivas et al. who studied two homozygous patients from two different families [60]. The RasGRP2 protein was not detectable in platelets from both homozygous patients. While in one patient platelet aggregation was almost absent in response to ADP (10 µM) or TRAP (25 µM) and significantly reduced in response to arachidonic acid (1.5 mM), the other patient had more sustained responses to all these agonists at the same doses (i.e., an intermediate response for ADP and PAR agonist and a normal response for arachidonic acid). Clot retraction can either be unaffected [58] or slightly impacted [59]. Flow cytometry using monoclonal antibodies directed against a range of membrane receptors does not reveal quantitative receptor deficiency, notably normal αIIbβ3 integrin surface expression. αIIbβ3 integrin activation by all agonists except PMA was impaired depending on the type of agonist and the dose used, the activation being defective upon stimulation with low doses and normal with high doses of agonists or with PMA. In accordance with a defect in Rap1 activation, granule secretion is reduced upon stimulation with low doses of an agonist [95,96]. Platelets from patients have a decreased ability to bind soluble and immobilized fibrinogen [58,59,60,95] and form thrombi over collagen at arterial shear rate [58,96], and exhibit a reduced number of filipodia and fail to form lamellipodia. Platelets enigmatically fail to spread on collagen under arterial flow [57].

As part of a differential diagnosis, the exceptional cases of variant-type of Glanzmann Thrombasthenia with normal αIIbβ3 surface expression but associated with a lack of platelet aggregation need to be excluded. Some patients with BDPLT18 were considered to be carriers of a variant form of Glanzmann thrombasthenia before whole exome sequencing unequivocally restored the diagnosis [84]. At this stage, sequencing of the *RASGRP2* gene will confirm the diagnosis and definitively exclude a variant form of Glanzmann thrombasthenia. 

### 7.4. RASGRP2 Gene Variations

BDPLT18 occurs worldwide; no geographical restriction of the disease appearance was noticed. Deficiency has been described in patients from European, Turkish, Jamaican, Argentinian, Japanese, Korean, and Chinese origins. It could be however more abundant in certain ethnic groups due to the high level of consanguinity within some communities.

The *RASGRP2* gene is highly polymorphic. Single nucleotide substitutions leading to nonsense or missense mutations, splicing defects, start codon loss, frameshifts, small deletions, and insertions are all common (Ensembl, release 98–September 2019) [97]. Most reported families have their own private mutation although some reoccur in unrelated families and identify gene “hotspots”. Twenty-two pathogenic variations have been reported so far (Figure 4 and Table 1). We add here another so far not described variant that affects the C1 domain of RasGRP2 at position 503. The variant was highlighted in a 14-year-old male that suffers severe bleedings that started in his early childhood. Bleeding symptoms are mainly recurrent epistaxis, spontaneous gum bleeding, excessive bleedings upon surgery, and dental extraction. The mutation corresponds to a homozygous G to T transition in exon 13 (c.1507G>T) that leads to the replacement of a glutamic acid residue to a stop codon (p.E503*) (Table 1, “referenced as newly identified variant”).

Among these twenty-three pathogenic variations, five were found in two or more families. Notably the p.Arg494Alafs*54 and the p.Phe497Serfs*22 are found in two and three unrelated families, respectively, either in an homozygous or in a compound heterozygous status [84,91]. Twelve of the described variations correspond to drastic modifications such as deletions, stop codon gains or changes of the reading frame (p.Asp25Ala*15, p.Asn67Leufs*24, p.Arg113*, p.Arg113Aspfs*6, p.Pro125*, p.Gln236*,p.Glu260*, p.Lys309*, p.Leu360del, p.Arg494Alafs*54, p.Phe497Serfs*22 and p.Glu503*). Of note is that two of these variants result from mutations in intronic regions (p.Asp25Ala*15 and p.Pro125*). The eleven other variations correspond to substitutions (p.Phe181Ser, p.Arg220Glu, p.Gly248Trp, p.Gly248Arg, p.Tyr289Cys, p.Cys296Tyr, p.Cys296Arg, p.Gly305Asp, p.Asn330Lys, p.Ala345Pro and p.Ser381Phe), all being localized in the CDC-25 homology domain suggesting that these mutations affect residues with strategic positions for RasGRP2 activity or stability. As an example, the p.Gly248Trp transition is the only variant described so far within the CDC-25 homology domain that does not induce a loss of RasGRP2 protein expression in platelets [58]. The mutation causes the substitution of a small neutral amino acid (glycine) by a large polar one (tryptophan) leading to a protrusion within a cavity of the GEF that interacts with Rap1. This modification is predicted to result in a less effective GDP to GTP exchange and a shift of Rap1 to its inactivated, GDP-bound state. Platelets from the p.Gly305Asp homozygous carriers express only residual levels of RasGRP2 protein in platelets [89] suggesting a putative role of this amino acid residue in the protein’s stability. Evaluation of RasGRP2 protein expression in platelets from other homozygous carriers (p.Asn67Leufs*24, p.Arg113*, p.Gln236*, p.Cys296Tyr, p.Ser381Phe) and from two compound heterozygous (p.Lys309*/ p.Leu360del and p.Glu260*/ p.Cys296Arg) revealed a total loss of the RasGRP2 protein expression (Table 1). A shortened RasGRP2 protein was detected in one homozygous p.Phe497Serfs*22 carrier [84]. Overall, these results indicate that BDPLT18 can be classified into quantitative or qualitative deficiencies. The measurement of intraplatelet RasGRP2 levels is of particular interest to provide a better description of the two types of BDPLT18 and to confirm the deleterious nature of novel gene variants that influence RasGRP2 platelet content. Interestingly, two recent publications reported the cases of patients suffering bleeding diathesis that carry *RASGRP2* variants associated with other mutations in the *P2RY12* and *FERMT3* genes. They code for the P2Y12 ADP receptor and kindlin3 respectively, both being involved in platelet function and αIIbβ3 integrin activation mechanisms. Remarkably, homozygous compound *RASGRP2* (p.Arg113Aspfs*6) and heterozygous *P2RY12* (p.Thr126fs*34) combined deficiencies resulted in more severe platelet aggregation defect and bleeding phenotype than those observed in homozygous P2Y12 and heterozygous RasGRP2 variants’ carriers [93]. *RASGRP2* and *FERMT3* genes are both located on chromosome 11q13.1. *FERMT3* deficiency results in leukocyte adhesion deficiency type III (LAD-III) that is the pathology characterized by severe platelet dysfunction and Glanzmann-thrombasthenia-like bleedings associated to hyperleukocytosis and immune deficiency and, inconstantly, osteopetrosis [98]. Interestingly, the combination of two homozygous variations in *RASGRP2* (p.Gly248Arg) and *FERMT3* (p.Ser40Leu) identified in the index case resulted in severe and recurrent bleedings but not by the immunological features classically noticed in LADIII patients [94]. Thus, the deleterious nature of this *FERMT3* variant on the expression and function of the encoded protein, kindlin 3, needs to be clearly established.

### 7.5. Patient Management

BDPLT18 requires specialist management which can be inspired by many of the recommendations established for Glanzmann thrombasthenia patients [99]. Indeed, the deficit in RasGRP2 leads to a loss of function of the αIIbβ3 integrin, as does Glanzmann thrombasthenia.

All people with such disorders should be registered with a reference center for hemostasis disorders with appropriate facilities for investigation and treatment, and 24/7 access. Affected individuals should also be issued with a card describing their condition, and it is advisable to give the patient and his/her primary care physician written information about the condition and its care as this disorder is uncommon and will mostly be unfamiliar to many medical staff. Advice should be given where necessary about lifestyle issues too (e.g., individuals with severe disorders should avoid contact and fall-risk sports) and patients should avoid medication which interferes with platelet function, (i.e., salicylates, NSAIDs and other antiplatelet agents). For women, the management of menorrhagia is essential because it is a major source of acute and chronic anemia, especially in teenage girls, and it has a strong impact on quality of life. In one young adolescent affected by BDPLT18, massive menorrhagia led to hemorrhagic shock that required red blood cell transfusion [96]. In another one, menorrhagia has been successfully controlled with oral contraceptives and tranexamic acid [59]. Anemia and martial deficiency secondary to bleeding episodes have been frequently reported in patients [58,59,84,91,94,96,100] and should be regularly detected and treated. Pregnancy should be managed in close collaboration with the specialized center in hemostasis, with a written management plan for the affected mother, and also a plan for investigation and management of the neonate, if necessary. We and others recently reported two cases of successful management of bleeding diathesis during the course of pregnancy and peripartum period in two women woman suffering BDPLT18 [90,100]. The neonate is not at risk of inheriting the full platelet function disorder unless the father is a carrier, although it should be noted that minor to no symptoms are seen in individuals who are heterozygous carriers of RasGRP2 variants. This may be important to consider because of the high rate of consanguinity observed in some communities. Screening of the father’s *RASGRP2* gene will identify carrier fathers and thus will help identifying neonates who are potentially at hemorrhagic risk.

As for Glanzmann thrombasthenia, in all BDPLT18 reported patients, bleeding complications have required medical intervention including antifibrinolytic treatment, transfusion of platelet or red blood cell concentrates [58,59,60,91,94,95] and occasionally desmopressin [60,96]. As an example, in the nine index cases we reported in the Westbury et al. study, seven of them (78%) have required at least one red cell or platelet transfusion indicating severe bleedings [91]. Treatment with antifibrinolytic agents, e.g., tranexamic acid, either orally, as a mouthwash or by intravenous injection may be considered in the case of moderate bleedings. This may prove to be useful to control menorrhagia and other mild bleeding manifestations from mucous membranes, such as epistaxis. Platelet transfusions are appropriate in cases of severe bleedings and when other agents have failed. However, these blood products carry risks of transfusion-transmitted infections and allergic reactions. Platelet and red blood cell transfusions should not be given without clear indications. Patients with BDPLT18 may be subject to repeated episodes of transfusion, putting them at risk of developing alloantibodies either against HLA or HPA antigens. However, the risk of developing isoantibodies is almost absent because αIIbβ3 integrin is normally expressed on BDPLT18 patient platelet surface, unlike Glanzmann thrombasthenia patients. Thus, the risk of neonatal thrombocytopenia should be lower. Nevertheless, in this population likely to frequently receive platelet concentrate transfusions, the search for alloantibodies before and after transfusions should be carried out to look for the presence of such antibodies and, in case of a positive test, to adapt the patient’s management. In Glanzmann thrombasthenia, rFVIIa (NovoSeven^®^) is preferred to platelet concentrates in the case of poor response to platelet transfusions, immunization against αIIbβ3 integrin or HLA, or when platelet concentrates are not readily available. Injections should be repeated every 2 to 3 h initially; progressive spacing is possible over a few days depending on circumstances and clinical course. The total number of injections required to treat bleeding episodes may vary from one patient to another and depending on the circumstances. Three injections must be made to achieve a hemostatic effect and before evaluating the possible failure of the treatment. We have reported the efficacy of rFVIIa treatment in a young woman with RasGRP2 deficiency during the postpartum period [100]. A preventive strategy with good efficacy consisted in the administration of tranexamic acid associated with platelet concentrates. She underwent an emergency hospitalization 38 days postpartum for a severe hemorrhage during the first postpregnancy menstrual period. Platelet and red blood cell transfusions, intravenous tranexamic acid and fluid infusions allowed hemodynamic stabilization but showed moderate hemostatic efficacy. A single rFVIIa injection (90 μg/kg) stopped abnormal bleeding. The patient left the hospital four days later. Thus, rFVIIa may be considered as a strategy to manage the hemorrhagic risk in women with BDPLT18 at delivery and during the following days. Recently, another 41-year-old Korean woman successfully delivered a healthy baby by Cesarian section. She was prophylactically transfused with two units of single-donor platelets before surgery and had only moderate blood loss (400 mL) during the surgery. She was then transfused with two units of leukocyte-reduced red blood cells and started an iron replacement therapy. She did not show any other medical issue during the one-month follow-up [90]. As in Glanzmann thrombasthenia, women need to be closely observed and tranexamic acid continued for at least several weeks and to have ready access to the obstetric service in connection with the hemostasis expert center.

Another patient was also successfully treated with rFVIIa alone during an hemorrhagic episode from unspecified origin [95]. In another case, rFVIIa was used during neurosurgery for meningioma removal due to the ineffectiveness of platelet transfusions [84]. Treatment was continued for 4.5 days. Blood loss was moderate and the postoperative period was without major complications. This same patient was also treated as a first-line treatment by rFVIIa during hernia repair surgery with good results. The patient’s sister was also successfully treated with rFVIIa to control bleeding during appendectomy in view of the ineffectiveness of platelet transfusions too [84].

Overall, management protocols of bleedings in BDPLT18 patients require the competences of multidisciplinary medical staff. Close monitoring and planned preventive haemostatic strategies are accordingly required to minimize bleedings in these high-risk patients. Even though transfusion protocols based on the use of conventional platelet concentrate are frequently used to treat or prevent bleedings, other therapeutic alternatives exist such as rFVIIa that has proven its efficacy in BDPLT18 patients.

## Figures and Tables

**Figure 1 ijms-21-01075-f001:**
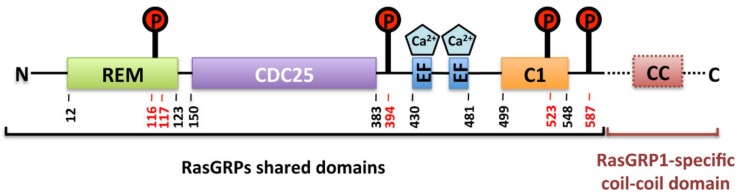
Structural organization of the Ras guanine-nucleotide releasing proteins (RasGRPs) with amino acide sequence annotation for RasGRP2. The protein domains indicated are the Ras exchange motif (REM), catalytic domain (CDC-25), calcium-binding helix(E)-loop-helix(F) hands (EF), diacylglycerol-binding domain (C1) and the RasGRP1 specific C-terminal coil-coil domain (CC). Circled P and red numbers correspond to phosphorylation sites on serine residues involved in RasGRP2 activity regulation.

**Figure 2 ijms-21-01075-f002:**
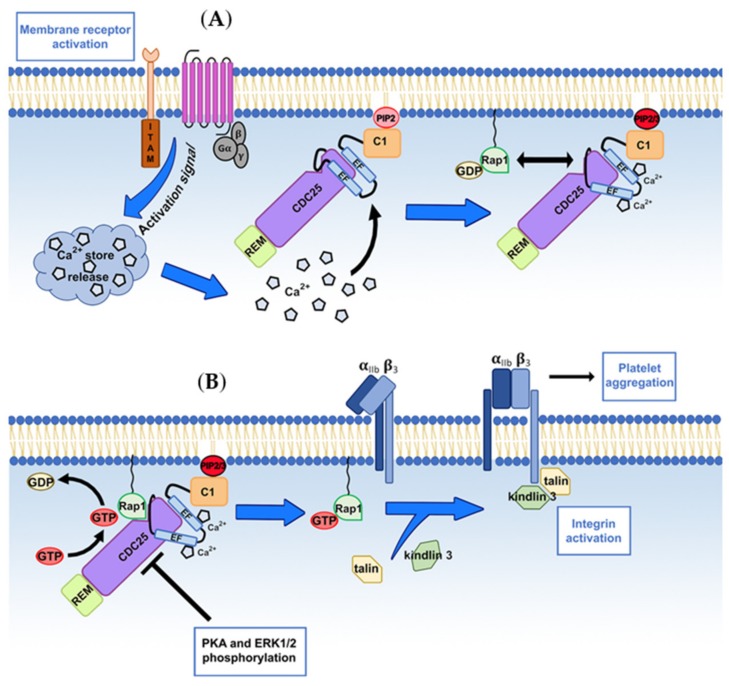
RasGRP2 activation mechanism and activity regulation during αIIbβ3-integrin “inside-out” signaling in platelets. (**A**) Platelet surface receptor activation by vascular adhesive proteins and/or soluble agonists initiates an intracellular activation signal that induces the release of Ca2+ from intracellular stores into the platelet cytoplasm. Ca2+ binding to the EF hands induces conformational changes that activate RasGRP2, located at the platelet membrane through the association of its C1 domain with the phosphoinositides PIP2 and PIP3. (**B**) The membrane-bound, activated RasGRP2 interacts with Rap1 at the proximity of the cell membrane, and facilitates GDP dissociation and its replacement by (guanosine triphosphate) GTP on the GTPase. The guanine-exchange activity of RasGRP2 can be controlled by PKA- and ERK1/2-dependent phosphorylations. The GTP-bound Rap1 favors the recruitment of talin and kindlin onto the β-chain of the αIIbβ3 integrin leading to its conformational change, activation and subsequent platelet aggregation.

**Figure 3 ijms-21-01075-f003:**
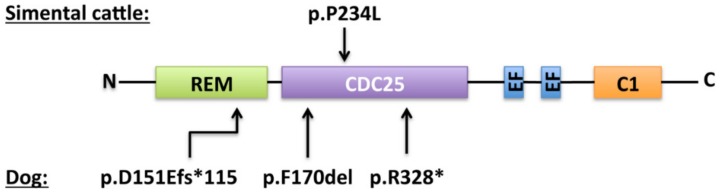
Localization and predicted consequence of the *RASGRP2* reported variants in Simmental cattle and dog. Sequences of the variants are annotated according to the consensus nomenclature to describe variant effect at the protein level (fs = frameshift, del = deletion, * = change to a stop codon).

**Figure 4 ijms-21-01075-f004:**
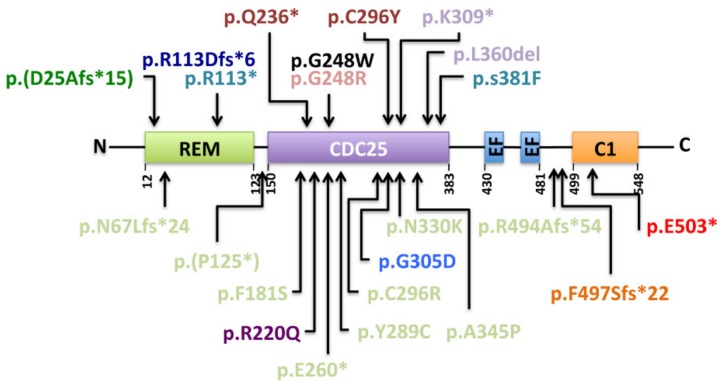
Localization and predicted consequence of the *RASGRP2* reported variants in humans. The variant reported by Canault et al. [58] is shown in black. The variants identified by Lozano et al. are highlighted in turquoise, by Kato et al. [96] in light purple, by Sevivas et al. [60] in brown, by Westbury et al. [91] in light green, by Desai et al. [84] and Westbury et al. in orange, by Bermejo et al. [89] and Wesbury et al. in light blue, Lyu et al. [92] in dark green, Yun et al. [90] in purple, Lunghi et al. [93] in dark blue, Manukjan et al. [94] in pink and the newly identified variant id indicated in red. Variants in brackets are prediction of intronic modifications affecting splice regions. Sequences of the variants are annotated according to the consensus nomenclature to describe variant effect at the protein level (fs = frameshift, del = deletion, * = change to a stop codon, variant in brackets represent intronic modification effects (splice variants))

**Table 1 ijms-21-01075-t001:** Characteristics and bleeding symptoms of the pathogenic *RASGRP2* variants reported to date.

Genomic Variation	Variant Type	Protein Effect	Sex	Platelet Expression	Age at Diagnosis	Age at Presentation	Bleeding Symptoms						Ref
							E	SC	M	DE	S	GI	
c.742G>T	Missense	p.Gly248Trp	F	Yes	55	1 year							[58]
			H	Yes	53	1 year							
			H	Yes	49	1 year							
c.337C>T	Stop codon	p.Arg113*	F	No	55	Lifelong							[59]
			M	No	46	Childhood							
c.1142C>T	Missense	p.Ser381Phe	M	No	9	lifelong							[59]
c.1142C>T c.659G>A	Missense Missense	p.Ser381Phe p.Arg220Glu	F	N/D	41	Early childhood							[90]
c.925A>T c.1081_1083delCTG	Stop codon Deletion	p.Lys309* p.Leu360del	F	No	16	Before 3 years							[89]
c.706C>T	Stop codon	p.Gln236*	M	No	8	1 year							[60]
c.887G>A	Missense	p.Cys296Tyr	F	No	4	First year							[60]
c.914G>A	Missense	p.Gly305Asp	M	No	9	Early childhood							[89,91]
			M	Yes (Residual)	24	2 years							
c.199delAA	Frameshift	p.Asn67Leufs*24	F	No	23	1 year							[91]
c.372-3C>G	Splice variant	p.(Pro125*)	F	N/D	24	2 years							[91]
c.990C>G	Missense	p.Asn330Lys	F	N/D	21	5 years							[91]
c.778G>T c.886T>C	Stop codon Missense	p.Glu260* p.Cys296Arg	F	No	20	1 year							[91]
c.1482InsG	Frameshift	p.Arg494Alafs*54	M	N/D	60	5 years							[91]
c.1482InsG c. 542T>C	Frameshift Missense	p.Arg494Alafs*54 p.Phe181Ser	M	N/D	13	1 year							[91]
c.1490delT	Frameshift	p.Phe497Serfs*22	M	Yes (Lower MW)	45	5 years							[84,91]
			F	Yes (Lower MW)	55	During the first year							
c.1490delT c.1033G>C	Frameshift Missense	p.Phe497Serfs*22 p.Ala345Pro	M	N/D	57	5 years							[91]
c.1490delT c.866A>G	Frameshift Missense	p.Phe497Serfs*22 p.Tyr289Cys	M	N/D	61	4 years							[91]
c.74-1G>C	Splice variant	p.(Asp25Ala*15)	M	N/D	9	3 years							[92]
c.337delC	Deletion	p.Arg113Aspfs*6	M	N/D	9	During the first year							[93]
c.742G>C	Missense	p.Gly248Arg	F	N/D	15	Early childhood							[94]
**Newly identified variant**													
c.1507G>T	Stop codon	p.Glu503*	M	N/D	14	Early childhood							
						**% of patients**	**96**	**84**	**73**	**40**	**40**	**16**	

Blue background corresponds to *RASGRP2* composite heterozygous variants; pink background corresponds to a *RASGRP2* variant associated to a *P2RY12* heterozygous variant; yellow background corresponds to a *RASGRP2* variant associated to a FERMT3 homozygous variant. E = Epistaxis, SC = Subcutaneous hemorrhage, M = Menorrhagia, DE = Prolonged bleeding after dental extraction, S = Prolonged bleeding after surgery and GI = Gastrointestinal hemorrhage.

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
