# Peer review of "RasGRP2 Structure, Function and Genetic Variants in Platelet Pathophysiology"

_ijms, 2020, doi:10.3390/ijms21031075_

Round 1

Reviewer 1 Report

In this paper the authors review the role that RasGRP2 plays in bleeding disorders and atherogenesis. Given the interest the role this protein plays in controlling platelet activation, this work detailing both the studies in knock-out mice as well as clinical studies on patients with bleeding disorders this work will likely be of interest to a number of people.  This review is generally well written bar a few minor grammatical errors and should be suitable for publication after some minor corrections outlined below.

Major comments

1. My only concern here is that although the authors do try to distinguish the species in which the platelet studies have been performed, this is not consistent throughout the whole piece.  I think it is important for readers to be able to judge for themselves the relevance to human (patho)physiology of studies performed in a mouse knock-out model.  This is particularly important for statements such as this: 

L127-L136 - "Interestingly, besides resulting in reduced platelet aggregation, minimal  expression of RasGRP2 (as low as ≈10% of expression in controls), strongly prevent arterial and immune complex-mediated thrombosis, but has only limited impact on primary hemostasis. Thus, RasGRP2 could represent a therapeutic target for the development of potentially safe
antithrombotic drugs with little impact on the bleeding risk"    

Given that therapeutic interventions found to work in animal models generally have a very low translation when tested in clinical studies (about 10%), and the significant differences in the known development of atherosclerotic plaques in mice, I would prefer the authors specified that the evidence referenced comes from mouse models.  

2. The Kd is reported as 100 nm (should be corrected to 100 nM). I think some context should be given to the reader as this would be a very sensitive system as this is very close to the resting platelet cytosolic calcium concentration of around 50 nM, and agonist stimulation in physiological conditions can cause the cytosolic Ca2+ concentration to rise to around 300 nM - 1 uM depending on agonist and dose.    

Minor comments

L83- reword this to remove the duplication of "posttranslational modifications" in the same sentence L109-110 IP3 induces release of Ca2+ from intracellular stores into the platelet cytosol  L111-112 "These Ca2+-sensitive and PKC pathways were shown [to] act separately but synergistically in the activation of αIIbβ3 integri  L128 - "...to form three-dimensional thrombi [when perfused] at arterial shear rate[s]

Author Response

Manuscript ID: IJMS-685320

Title: “RasGRP2 structure, function and genetic variants in platelet pathophysiology.”

Dear Editor Wang,

We would like to thank you and the referees for their comments.

In the revised version of the manuscript, we have specifically addressed the issues raised by both reviewers. All corrections and modification in the manuscript are highlighted in red.

We hope that our manuscript is now suitable for publication in the International Journal for Molecular Sciences.

Below you will find our answers to the reviewer 1 comments.

Reviewer 1:

Major comments

My only concern here is that although the authors do try to distinguish the species in which the platelet studies have been performed, this is not consistent throughout the whole piece.  I think it is important for readers to be able to judge for themselves the relevance to human (patho)physiology of studies performed in a mouse knock-out model.  This is particularly important for statements such as this: 

L127-L136 - "Interestingly, besides resulting in reduced platelet aggregation, minimal  expression of RasGRP2 (as low as ≈10% of expression in controls), strongly prevent arterial and immune complex-mediated thrombosis, but has only limited impact on primary hemostasis. Thus, RasGRP2 could represent a therapeutic target for the development of potentially safe
antithrombotic drugs with little impact on the bleeding risk"    

Given that therapeutic interventions found to work in animal models generally have a very low translation when tested in clinical studies (about 10%), and the significant differences in the known development of atherosclerotic plaques in mice, I would prefer the authors specified that the evidence referenced comes from mouse models.  

We thank Reviewer 1 for his comment. We fully agree that it is essential to specify in which species the platelet studies were conducted. We have therefore modified the manuscript where necessary.

L127  : Consistent with the involvement of RasGRP2 in αIIbβ3 integrin activation, mouse platelets lacking the GEF show a markedly reduced ability to form three-dimensional thrombi when perfused at arterial shear rates both in vitro and in vivo and dramatically prolonged bleeding time [11], [41].

L132: Interestingly, hypomorphic mice expressing minimal levels of human RasGRP2 instead of endogenous RasGRP2 ( ≈10% of expression in controls) showed reduced platelet aggregation and severely impaired arterial and immune complex-mediated thrombosis with only slightly affected primary hemostasis [53].

L138: In parallel, RasGRP2 also contributes to thromboxane A2 generation and release from mouse platelet, thus reinforcing the second wave of platelet activation signal through PKC-mediated ADP secretion and the P2Y12/PI3K-dependent RASA3 inhibition pathway [48], [54].

L153: The signaling module RasGRP2/Rap1 was also shown to play an important role in the conversion of platelets to a pro-coagulant state since RasGRP2-deficiency leads to impaired phosphatidylserine exposure on mouse platelet surface and delayed and reduced fibrin generation at the vascular lesion site [61].

L158: Indeed, RasGRP2/Rap1-dependent signal promotes atherosclerotic plaque formation in mice and determines its composition probably through platelet activation and platelet-leukocyte aggregate formation. However, but RasGRP2 is also expressed in leukocytes, thus the exact contribution of platelet- and/or leukocyte-associated RasGRP2-dependent signal in atherosclerosis still remains elusive [62].

L166: Similarly, RasGRP2 enhances the adhesion ability of human T cells through lymphocyte function-associated antigen-1 (LFA1) and contributes to the interaction with intercellular adhesion molecule-1 (ICAM-1) [14].

L196: RasGRP2 was initially identified as a protein with enriched expression in human and rodent brain basal ganglia and in their axon-terminal regions [7], [73].

The Kd is reported as 100 nm (should be corrected to 100 nM). I think some context should be given to the reader as this would be a very sensitive system as this is very close to the resting platelet cytosolic calcium concentration of around 50 nM, and agonist stimulation in physiological conditions can cause the cytosolic Ca2+ concentration to rise to around 300 nM - 1 uM depending on agonist and dose.    

L214 : The “nanomolar” symbol (nM) was corrected. In agreement with the reviewer’s comment, we added a sentence indicating the high sensitivity of RasGRP2 to cytoplasmic calcium concentration changes and indicated the calcium concentration ranges in resting and activated platelet cytoplasm.

RasGRP2 binds to calcium via its two EF hand domains which have a high affinity for calcium (Kd < 100 nM) [77], making RasGRP2 extremely sensitive to activation, since the concentration of cytoplasmic calcium ranges from 25-100 nM in resting platelets and can increase up to micromolar levels upon activation depending on the agonist and the dose [78]. The binding of calcium to the EF hand domains results in major conformational changes [25].

Minor comments

L83- reword this to remove the duplication of "posttranslational modifications" in the same sentence

L109-110 IP3 induces release of Ca2+ from intracellular stores into the platelet cytosol  

L111-112 "These Ca2+-sensitive and PKC pathways were shown [to] act separately but synergistically in the activation of αIIbβ3 integri  L128 - "...to form three-dimensional thrombi [when perfused] at arterial shear rate[s] 

We thank Reviewer 1 for noticing these typo mistakes. We modified the manuscript text accordingly.

L83: The canonical isoform 1 of RasGRP2 (Q7LDG7-1) is a 609 amino acid long (69.25 kDa) protein that possesses various post-translational modification sites for posttranslational modifications that were identified through high-throughput proteomic analyses (data obtained from PhosphoSitePlus [31]) and could affect the activity and/or the fate of the GEF.

L109: IP3 induces the release of Ca2+ from intracellular stores into the platelet cytoplasm [37], [38] and DAG activates protein kinases C (PKCs) that results in platelet sustained granule secretion, subsequent ADP release and P2Y12 receptor activation.

L111: These Ca2+-sensitive and PKC pathways were shown to act separately but synergistically in the activation of αIIbβ3 integrin [39], [40]. 

Reviewer 2 Report

This is a comprehensive and on the whole well-written review of RasGRP2 structure and function describing how its mutation results in a platelet bleeding diathesis. The english expression is in places awkward and there are minor grammatical and other errors that need to be carefully edited. The manuscript would be strengthened by a figure showing the association and regulation of RasGRP2 function with respect to its role in the integrin inside-out signalosome.

Author Response

Manuscript ID: IJMS-685320

Title: “RasGRP2 structure, function and genetic variants in platelet pathophysiology.”

Dear Editor Wang,

We would like to thank you and the referees for their comments.

In the revised version of the manuscript, we have specifically addressed the issues raised by both reviewers. All corrections and modification in the manuscript are highlighted in red.

We hope that our manuscript is now suitable for publication in the International Journal for Molecular Sciences.

Below you will find our answers to the reviewers' comments.

Reviewer 2:

This is a comprehensive and on the whole well-written review of RasGRP2 structure and function describing how its mutation results in a platelet bleeding diathesis. The english expression is in places awkward and there are minor grammatical and other errors that need to be carefully edited. The manuscript would be strengthened by a figure showing the association and regulation of RasGRP2 function with respect to its role in the integrin inside-out signalosome.

We thank Reviewer 2 for his/her comments. We have added a figure that summarizes the activation and activity regulation mechanisms of RasGRP2 by integrating them with the "inside-out" signaling of the integrins. This figure has now been added as a new Figure 2. We have also done our best to improve the English expression throughout the manuscript. If requested by Reviewer 2 and/or the Editor, we are up to have our manuscript corrected by an English editing service because we don't want our review to sound awkward.